# A Curious Case of Remarkable Resilience to Gradient Attacks via Fully Convolutional and Differentiable Front End with a Skip Connection

**Leonid Boytsov**[*]                                                                *leo@boytsov.info*
*Bosch Center for Artificial Intelligence, Pittsburgh, USA*

**Ameya Joshi**[**]
*Bosch Center for Artificial Intelligence, Pittsburgh, USA*

**Filipe Condessa**[**]
*Bosch Center for Artificial Intelligence, Pittsburgh, USA*

**Reviewed on OpenReview:** *https://openreview.net/forum?id=kt7Am2wHlm*

## Abstract

We experimented with front-end enhanced neural models where a differentiable and fully convolutional model with a skip connection added before a frozen backbone classifier. By training such composite models using a small learning rate for about one epoch, we obtained models that retained the accuracy of the backbone classifier while being unusually resistant to gradient attacks—including APGD and FAB-T attacks from the AutoAttack package—which we attribute to gradient masking.

Although gradient masking is not new, the degree we observe is striking for fully differentiable models without obvious gradient-shattering—e.g., JPEG compression—or gradient-diminishing components. The training recipe to produce such models is also remarkably stable and reproducible: We applied it to three datasets (CIFAR10, CIFAR100, and ImageNet) and several modern architectures (including vision Transformers) without a single failure case.

While black-box attacks such as the SQUARE attack and zero-order PGD can partially overcome gradient masking, these attacks are easily defeated by simple randomized ensembles. We estimate that these ensembles achieve near-SOTA AutoAttack accuracy on CIFAR10, CIFAR100, and ImageNet (while retaining almost all clean accuracy of the original classifiers) despite having near-zero accuracy under adaptive attacks.

Moreover, adversarially training the backbone further amplifies this front-end "robustness". On CIFAR10, the respective randomized ensemble achieved 90.8±2.5% (99% CI) accuracy under the full AutoAttack while having only 18.2±3.6% accuracy under the adaptive attack ($\varepsilon = 8/255$, $L^\infty$ norm). While our primary goal is to expose weaknesses of the AutoAttack package—rather than to propose a new defense or establish SOTA in adversarial robustness—we nevertheless conclude the paper with a discussion of whether randomized ensembling can serve as a practical defense.

Code and instructions to reproduce key results are available. `https://github.com/searchivarius/curious_case_of_gradient_masking`

---

[*]Work done outside the scope of Amazon employment, primarily while employed by Bosch.
[**]Work completed while employed by Bosch.

# 1  Introduction

In the field of adversarial machine learning, the AutoAttack package—henceforth simply AutoAttack—has gained considerable popularity as a standard approach to evaluate model robustness. According to Google Scholar, mid 2025 core AutoAttack papers (Croce & Hein, 2020a;b; Andriushchenko et al., 2020) had been cited about 4400 times. AutoAttack is the "engine" of the RobustBench benchmark Croce et al. (2021), which is a popular benchmark tracking progress in adversarial machine learning. However, the reliability of AutoAttack itself, as well as that of its individual "components," has received little to no discussion in the literature. Tramèr et al. (2020) noted in passing that AutoAttack overestimated robust accuracy of several models, which had nearly zero accuracy under adaptive attacks, but they did not discuss the actual failure modes of AutoAttack.

To partially address this gap, we have carried out a case study where AutoAttack is applied to front-end "enhanced" models. We have found such composite models to be surprisingly resistant to standard projected gradient (PGD) attacks as well as to gradient attacks from AutoAttack (Croce & Hein, 2020a). We attribute this behavior to gradient masking (Athalye et al., 2018; Tramèr et al., 2020) and provide supporting evidence. Interestingly, the FAB attack from AutoAttack was previously shown to be resilient to gradient masking (Croce & Hein, 2020b). Yet, we have found it to be ineffective as well.

Despite the fact that RobustBench does not accept randomized models (Croce et al., 2021), users may apply AutoAttack to randomized defenses unaware that AutoAttack can fail against them (in fact, this was our experience). Moreover, AutoAttack was shown to be effective against complex randomized defenses (Croce & Hein, 2020a). Thus, it was reasonable to expect AutoAttack to be successful against a simple randomized ensemble as well. When AutoAttack detects a randomized defense it only produces an easy-to-miss warning and suggests using the randomized version of the attack. However, the randomized version of AutoAttack has only gradient attacks, which—we find—can be *ineffective* in the presence of gradient masking.

Even though the gradient masking phenomenon is not new, the degree of this masking is quite remarkable for fully differentiable models that do not have gradient-shattering components such as JPEG compression or components that are expected to cause diminishing gradients (we also carried out sanity checks to ensure there were no numerical anomalies, see § A.2). In particular, the models retain full accuracy on original (non-adversarial) data, but a "basic" 50-step PGD with 5 restarts degrades performance of a front-end enhanced model by only a few percent. At the same time it is possible to circumvent gradient masking with a nearly 100% success rate using a variant of the backward-pass differentiable approximation Athalye et al. 2018.

Although black-box attacks such as the SQUARE attack and the zero-order PGD can be effective against gradient masking, these attacks are easily defeated by combining gradient-masking models into simple randomized ensembles. We estimate that these ensembles achieve near-SOTA AutoAttack accuracy on CIFAR10, CIFAR100, and ImageNet despite having virtually zero accuracy under adaptive attacks. Quite interestingly, adversarial training of the backbone classifier can further increase the resistance of the front-end enhanced model to gradient attacks. On CIFAR10, the respective randomized ensemble achieved 90.8±2.5% accuracy under AutoAttack (99% confidence interval).

We do not aim to establish SOTA in adversarial robustness. Our primary goal is to expose weaknesses of AutoAttack. In that, our results further support the argument that adaptive attacks designed with the *complete* knowledge of model architecture are crucial in demonstrating model robustness (Tramèr et al., 2020). We nevertheless believe that automatic evaluation is a useful baseline and we hope that our findings can help improve robustness and reliability of such evaluations. Beyond exposing weaknesses of AutoAttack, we also examine whether randomized ensembling can serve as a practical defense, and conclude the paper with a discussion of its potential. Code and instructions to reproduce key results are available.[1]

# 2  Related Work

Our paper focuses on a specific sub-field concerned with test-time attacks and defenses for neural-network classifiers on image data, which dates back only to 2013 (Szegedy et al., 2014; Biggio et al., 2013). For a

---

[1] https://github.com/searchivarius/curious_case_of_gradient_masking

detailed overview, we address the reader to recent surveys (Chakraborty et al., 2021; Akhtar et al., 2021). However, a broader field of adversarial machine learning has a longer history with earlier applications such as spam and intrusion detection (Dalvi et al., 2004; Lowd & Meek, 2005; Biggio & Roli, 2018).

Originally, most practical classifiers were linear and despite successes in attacking them there was a belief that attacking complex and non-linear models would be difficult (Biggio & Roli, 2018). This assumption was partly rooted in the intuition that increased model complexity could make decision boundaries harder to approximate or exploit. However, neural networks—while non-linear—turned out to be vulnerable in ways similar to or even more so than their linear counterparts (Biggio et al., 2013; Szegedy et al., 2014). As Biggio & Roli (2018) noted, adversarial machine learning gained quite a bit of momentum after the publication by Szegedy et al. (2014). In addition to showing that deep neural networks were vulnerable to small adversarial perturbations, Szegedy et al. (2014) found that these perturbations were (to a large extent) *transferable* among models.

Since then, the field has been growing exponentially, with more than twelve thousand papers uploaded to arXiv from the beginning of 2018 to mid-2025 (Carlini, 2019). Despite such explosive growth, adversarial training is arguably the only reliable defense. Nevertheless, it does *not* provide theoretical guarantees. Some assurances can be provided by provable defenses, but they have limited applicability (Croce & Hein, 2020a; Akhtar et al., 2021).

Quite a few additional empirical (sometimes sophisticated) defenses have been proposed, but they have been routinely broken using strong adaptive attacks (Athalye et al., 2018; Tramèr et al., 2020). This, in turn, motivated development of automatic evaluation suites, which can reduce manual effort often required to execute adaptive attacks. Among them, AutoAttack (Croce & Hein, 2020b) is arguably the most popular package, which includes several hyperparameter-free gradient attacks as well as a query-efficient black-box SQUARE attack (Andriushchenko et al., 2020). About one year later, Croce et al. (2021) created the RobustBench benchmark to track progress in adversarial machine learning where adversarial robustness is measured using the accuracy under AutoAttack.

Although Croce & Hein (2020a) advertised AutoAttack as a reliable attack, Tramèr et al. (2020) noted in passing that AutoAttack overestimated robust accuracy of models in several cases, but without providing details. In that, research on weaknesses of AutoAttack appears to be scarce. For example, Lorenz et al. (2021) question the general transferability of RobustBench findings to more realistic settings, but not reliability of the attack itself. In particular, our finding regarding extreme gradient masking appears to be unknown. Methodologically, the most related paper is by Tramèr et al. (2020), who emphasized the importance of adaptive attacks. The main difference is that we focus on documenting one specific use of (extreme) gradient masking and simple randomization.

Note that gradient masking is typically associated with gradient shattering via non-differentiable components and randomization as well as components encouraging vanishing gradients (Athalye et al., 2018), but not with fully differentiable components with skip connections such as the denoising front end used in our study (see Figure 1 in § 4.1). We found only one case in which a fully differentiable convolutional network became more resistant to PGD attacks due to a special training procedure, namely, training with differential privacy (Boenisch et al., 2021; Tursynbek et al., 2020). However, the seemingly increased robustness was observed only on MNIST (LeCun et al., 2010) and in the low accuracy zone (see Fig. 1a in the paper by Boenisch et al. 2021). In contrast, in our case gradient-masked models achieve an accuracy of about 80-90% under a standard PGD attack while remaining fully breakable (see Table 1).

Randomized defenses fall under the broader category of moving-target defenses (Evans et al., 2011). Although our paper focuses on the weaknesses of AutoAttack rather than on proposing a new defense through randomization or obfuscation, it is worth discussing several typical approaches. This discussion covers some key approaches, but it is not intended to be exhaustive. Randomization and obfuscation broadly aim to prevent attackers from reproducing the exact data or model conditions used at prediction time. Existing approaches fall into two broad categories: the first involves random or pseudo-random modifications to the input data, while the second randomizes the model architecture, its parameters, or a combination thereof.

For example, Taran et al. (2019) use an ensemble of models, each operating on differently perturbed input data. A perturbation is created by (1) first transforming input using an invertible parameterized transformation $W_i$, (2) modifying the result using another parameterized transform $P_k$, and (3) then mapping it back using the inverse transformation $W_i^{-1}$. The choice of transformations is determined by secret keys unknown to the attacker. However, there is no inference-time randomization and the defense can potentially be circumvented by black-box attacks such as the SQUARE attack (Andriushchenko et al., 2020). In contrast, Qin et al. (2021) proposed to add Gaussian noise before passing inputs to the model—a method that aims to circumvent black-box attacks. Randomized smoothing also relies on randomly perturbing input data multiple times. It uses majority voting over noisy predictions to create a robust classifier, which also provides certification (Cohen et al., 2019).[2]

Model randomization includes a variety of defense strategies, with randomized ensembles being one of the simplest and most natural. In this approach, a subset of models (possibly just a single one) is selected at inference time to make a prediction. Even if the attacker has access to all models in the ensemble, they cannot know in advance which subset will be used for a given input. While we do not know what the first paper is that formally introduced randomized ensembles as a defense, Sengupta et al. (2019) proposed a game-theoretic extension of this idea, where optimal selection probabilities are computed based on each model's performance on benign and adversarial inputs. A key limitation of their analysis is the assumption that the adversary generates attacks using only one model at a time. In practice, it may be possible to construct adversarial examples that transfer across all models—an attack strategy not considered in their work. For example, one can generate adversarial samples by attacking a model ensemble (Liu et al., 2017; Gubri et al., 2022).

That said, it is possible to reduce transferability among ensemble models. For example, Bui et al. (2021) proposed a method that combines minimizing the cross-entropy loss for one model in the ensemble on an adversarial example with maximizing the loss for other models that misclassify the same input. Importantly, it is not always the same model whose loss is maximized—each adversarial example is associated with a different weighting of loss components, depending on whether each model predicts it correctly. Although the additional maximization term discourages correct predictions, it also encourages mispredicting models to produce highly uncertain outputs, with post-softmax probabilities approaching a uniform distribution. As a result, the influence of such mispredictions on the ensemble's overall prediction remains limited (the prediction is based on summing logits of all models in the ensemble).

Sharif et al. (2019) proposed a variant of adversarial training in which adversarial examples are generated using targeted attacks. Instead of searching for arbitrary perturbations that maximize the cross-entropy loss for the correct label $y$, they define a random label mapping $m(y)$—called a *derangement*—such that $m(y) \neq y$ for all $y$. In other words, $m$ is a randomly selected permutation that never maps a label to itself. Given an input $x$ with label $y$, they then construct adversarial examples that minimize the cross-entropy loss with respect to the target label $m(y)$, which is different from true label $y$. To promote diversity within the ensemble, a different derangement is sampled for each model.

Although not suggested for reducing intra-model transferability, Sitawarin et al. (2024) proposed light adversarial training where a target model is trained using adversarial examples generated for a set of publicly available models. This can also be applied to a multi-model scenario. To this end, at each training step one can sample a pair of models $A$ and $B$. Then, they generate an attack using the model $A$ and apply it to the adversarial training of model $B$.

Bai et al. (2024) observed that attacks exhibit limited transferability between an adversarially trained model and a model trained on original (clean) data—a finding we also confirm in our study (see § A.1). They exploited this insight by linearly combining the logits of the robust and standard classifiers (without retraining) and then applying a *non-differentiable* nonlinear transformation (with three parameters) to the result. The parameters of this transformation are optimized to maximize robustness.

---

[2]Certification is a formal guarantee that the classifier's prediction does not change within a certain neighborhood, defined by a certification radius $R$. In other words, it ensures that the prediction is unaffected by any adversarial perturbation with radius at most $R$.

A disadvantage of random ensembling is that it requires training and hosting of multiple models. A more efficient alternative is self-ensembling, which introduces randomness directly into a single model during training and/or inference. A basic example is test-time dropout, which—as we found—despite its simplicity can fool the SQUARE attack. It is also worth noting that randomization of input data with subsequent application of the same model—e.g., as in the randomized smoothing (Cohen et al., 2019)—can be seen as a form of self-ensembling as well.

A number of more sophisticated self-ensembling techniques have also been proposed. Liu et al. (2018) add a random noise layer before each convolutional layer, injecting Gaussian perturbations into the layer's input. Dong et al. (2022) found that using different normalization approaches (e.g., batch normalization vs. instance normalization) reduces transferability of attacks, and they propose a random normalization aggregation approach, which randomly selects the normalization method for each layer that uses normalization.

Fu et al. (2021) proposed to train models using randomly selected precision. They combine random-precision inference (RPI) and random-precision training (RPT): RPI selects inference precision at run-time to exploit low attack transferability across precisions, while RPT randomly switches training precisions with switchable batch normalization to further reduce attack transferability and enhance robustness.

## 3 Background and Notation

### 3.1 Preliminaries

We consider a $c$-class image classification problem with a neural network $h_\theta(x)$, where $\theta$ is a vector of network parameters. We will also use $f_\theta(x)$ to denote a special front-end network, which is optionally prepended to a standard vision classification backbone such as ResNet (He et al., 2016). Each input image is a vector $x$ of dimensionality $d = w \cdot h \cdot 3$ (width $\times$ height $\times$ the number of color channels), which "resides" in a normed vector space. In our work we use exclusively the $L^\infty$ norm and use $\mathcal{B}_\varepsilon(x)$ to denote an $L^\infty$ ball of the radius $\varepsilon$ centered at $x$.

Given input $x$, the network estimates class-membership probabilities of input $x$ by producing a vector of dimensionality $c$ whose elements are called soft labels:

$$h_\theta(x) : \mathbb{R}^d \to [0, 1]^c$$

The actual prediction (a hard label) is taken to be the class with the highest probability:

$$\arg\max_i h_\theta(x)_i$$

For training purposes the accuracy of prediction is measured using a loss function $L$. The value of the loss function $L$ for input $x$ and true output $y$ is denoted as:

$$L(h_\theta(x), y)$$

In this work—as it is common for classification problems—we use exclusively the cross-entropy loss.

### 3.2 Adversarial Example and Threat Models

Adversarial examples are *intentionally* (and typically *inconspicuously*) modified inputs designed to fool a machine learning model. In the untargeted attack scenario, the adversary does not care about specific class outcomes as long as the model can be tricked to make an incorrect prediction. In the targeted attack scenario, an attacker's objective is to make the model predict a specific class, e.g., to classify all images of pedestrians as motorcycles. The original unmodified inputs are often termed as clean data. The accuracy of the model on clean data is typically termed as *clean* accuracy. In contrast, the accuracy of the model on the adversarial examples is termed as *robust* accuracy.

In this paper, we consider adversarial input perturbations whose $L^\infty$ norm does not exceed a dataset-specific threshold $\varepsilon$. In other words, adversarial examples must be inside the ball $\mathcal{B}_\varepsilon(x)$, where $x$ is the original input.

Different types of attacks require different levels of knowledge about the model and its test time behavior. The attacker can query the model (possibly a limited number of times) by passing an input $x$ through the model and measuring one or more outcomes such as:

- model prediction $\arg\max_i h_\theta(x)_i$;
- soft labels $h_\theta(x)$;
- the value of the loss $L(h_\theta(x), y)$;
- the gradient with respect to input data $\nabla_x L(h_\theta(x), y)$.

In some scenarios, the attacker can additionally know the model architecture and weights. However, in the case of randomized models, it would not be possible to know test-time values of random parameters. In our study, we consider three levels of access:

- *black-box* attacks, which can obtain model predictions $\arg\max_i h_\theta(x)_i$ or the full soft label distribution $h_\theta(x)$. In the latter case, the attacker can also compute the loss value $L(h_\theta(x), y)$. Black-box attacks are commonly referred to as query-based attacks.;
- *gradient* attacks, which can *additionally* access gradients $\nabla_x L(h_\theta(x), y)$;[3]
- *full train-time access* attacks, which can *additionally* "know" all the training-time parameters such as model architecture and weights, but not the test-time random parameters.

### 3.3 Standard and Adversarial Training

Given a classifier $h_\theta(x)$ the standard training procedure aims to minimize the empirical risk on the *clean*, i.e., original training set $\mathcal{D}$:

$$\min_\theta \sum_{x,y \in \mathcal{D}} L(h_\theta(x), y) \tag{1}$$

To train models resistant to adversarial attacks Szegedy et al. (2014) and Goodfellow et al. (2015) proposed to expand or replace the original training set with adversarial examples generated by the model. This general idea has several concrete realizations, which are broadly termed as *adversarial training*. As noted by Goodfellow et al. (2015), "the adversarial training procedure can be seen as minimizing the worst-case error when the data is perturbed by an adversary." Using a popular formalization of this statement due to Madry et al. (2018), an adversarial training objective can be written as the following minimax optimization problem:

$$\min_\theta \sum_{x,y \in \mathcal{D}} \max_{\hat{x} \in \mathcal{B}_\varepsilon(x)} L(h_\theta(\hat{x}), y) \tag{2}$$

The inner maximization problem in Equation 2 corresponds to generation of adversarial examples for a given data point and a model. This problem is generally intractable: In practice, it is solved approximately using methods such as projected gradient descent (Biggio et al., 2013; Madry et al., 2018). In Section 4.2 we describe several approaches for finding adversarial examples (including PGD), which are used in our experiments.

---

[3]Note that these gradients are computed with respect to an input image, rather than with respect to model parameters. Indeed, the objective of the attack is to find the input maximizing the loss rather than the model parameters (that minimize the loss).

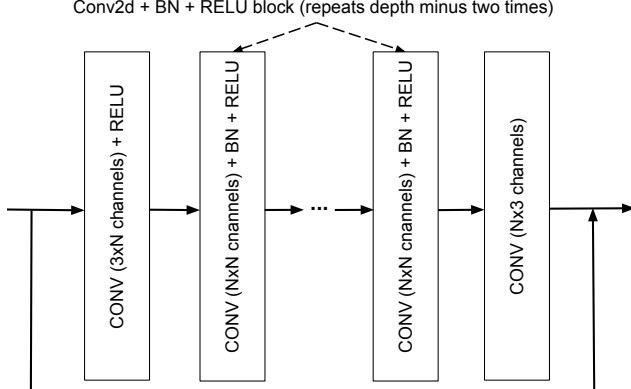

Figure 1: Denoising front end (DnCNN) architecture proposed by Zhang et al. (2017)): Note the skip connection.

## 4 Models and Attacks

### 4.1 Model Architectures and Training

In our experiments we use several types of models:

- Standard (non-robust) models, i.e., models trained using a standard objective (see Equation 1);
- Random ensembles of three standard/non-robust models;
- A pre-trained classifier is combined with a front end, and the resulting model is further trained using a modified variant of adversarial training, with the backbone frozen. The backbone can be either robust or standard (i.e., non-robust);
- Random ensembles of three models with front ends;
- Models trained using a traditional (multi-epoch) variant of adversarial training (see Equation 2).

In that we use three types of popular backbone models:

- Standard ResNet models of varying depth (He et al., 2016);
- Wide ResNet with depth 28 and widening factor 10 (Zagoruyko & Komodakis, 2016);
- ViT with varying patch sizes (Dosovitskiy et al., 2021);

A random ensemble is simply a set of three models, which are selected randomly and uniformly each time we obtain predictions $h_\theta(x)$ or compute gradients $\nabla_x L(h_\theta(x), y)$. It is noteworthy that randomization may unintentionally occur in PyTorch models that are inadvertently left in the training mode. In such a case, the model behaves similarly to a randomized ensemble due to enabled test-time use of dropout (Srivastava et al., 2014).

In the case of front-end enhanced models, an input image is first passed through a pre-processing network $f_{\theta_1}(x)$, also referred to as the front end. Following Norouzzadeh et al. (2021), we use a popular Gaussian Denoiser model DnCNN (Zhang et al., 2017) to process the image. DnCNN is a fully-convolutional neural network with a skip connection (see Figure 1). The rationale behind this architecture is to learn a residual mapping: instead of directly predicting the clean image, the network learns to estimate the noise component. This enables the model to approximate an identity transformation for clean regions while selectively correcting corrupted regions. Such a residual formulation simplifies the learning problem and accelerates convergence, especially when the corruption is relatively small.

The front end is prepended to a standard vision backend such as ResNet (He et al., 2016) or ViT (Dosovitskiy et al., 2021), which we denote as $h_{\theta_2}(x)$. The full network is represented by a composite function $h_{\theta_2}(f_{\theta_1}(x))$.

Unlike many other front-end defenses (Athalye et al., 2018), DnCNN does not explicitly shatter or obfuscate gradients. Moreover, DnCNN is a fully differentiable model with a skip connection, which is commonly used to improve gradient flow (He et al., 2016). Note that we use DnCNN without changing its architecture.

We train the front-end enhanced model using adversarial training (see Equation 2) with a "frozen" backbone model: This means that only parameters of the front end $f_{\theta_1}(x)$ are modified (using PGD to generate adversarial examples). In the case of the ResNet (and WideResNet) backends (He et al., 2016; Zagoruyko & Komodakis, 2016), we disable updates of the batch-normalization layers (Ioffe & Szegedy, 2015). Although our training procedure is somewhat similar to the approach of Norouzzadeh et al. (2021), there are two crucial differences:

- We use the original adversarial training objective *without* additional denoiser-specific losses;

- We use an extremely small (on the order of $10^{-6}$) learning rate and train for at most one epoch. Increasing the learning rate or extending training time does not yield a model that both resists strong gradient attacks and retains the clean model's accuracy. In contrast, training a strong adversarial model on CIFAR datasets typically requires dozens of epochs.

Our training procedure also differs from that of the original DnCNN, which was trained in a fully supervised manner to remove noise. Specifically, Zhang et al. (2017) added random noise to clean images and trained the denoiser to reconstruct the original clean versions.

As mentioned above, several models in our work are trained using a more conventional adversarial training approach. In this case, there is no front end and no parameter freezing. These models are trained for dozens of epochs using learning rates and schedules similar to those used in standard training.

## 4.2 Attacks

Gradient attacks used in our work require access to the loss function $L(h_\theta(x), y)$, which is used in a slightly different way for targeted and untargeted attacks. In the case of an untargeted attack, the adversary needs to find a point $\hat{x}$ (in the vicinity of the clean input $x$) with the smallest "likelihood" of the correct class $y$. This entails solving the following constrained maximization problem:

$$\hat{x} = \arg\max_{x \in \mathcal{B}_\varepsilon(x)} L(h_\theta(x), y) \tag{3}$$

In the case of a targeted attack, the adversary "nudges" the model towards predicting a desired class $\hat{y}$ instead of the correct one $y$. Thus, they minimize the loss of a class $\hat{y}$. This requires solving the following constrained minimization problem:

$$\hat{x} = \arg\min_{x \in \mathcal{B}_\varepsilon(x)} L(h_\theta(x), \hat{y}) \tag{4}$$

There are situations in which the model's loss surface or computational pathway causes gradient-based adversarial attacks to appear ineffective—not because the model is genuinely robust, but because the gradients are misleading, noisy, or uninformative. This phenomenon is commonly referred to as *gradient masking* or *gradient obfuscation* (Athalye et al., 2018; Tramèr et al., 2020).

There are different approaches to solving Equation 3 or Equation 4. Most commonly, they include first-order gradient-based methods, their zero-order approximations, and random search. In what follows we briefly survey attacks used in our work. For simplicity of exposition, we focus on the *untargeted* attack case.

**Projected Gradient Descent (PGD).** Projected Gradient Descent (Biggio et al., 2013; Madry et al., 2018) is an iterative optimization procedure, where in each iteration we make a small step in the direction of the loss gradient and project the result onto $\mathcal{B}_\varepsilon(x)$ to make sure it remains sufficiently close to the initial clean example:

$$x_{n+1} = \text{proj}_{\mathcal{B}_\varepsilon(x)} \left( x_n + \alpha \cdot \text{sign} \nabla_{\mathbf{x}} L(h_\theta(x), \hat{y}) \right), \tag{5}$$

where $\text{sign}(x)$ is an element-wise sign operator and $\alpha$ is a step size. PGD with restarts repeats this iterative procedure multiple times starting from different random points inside $\mathcal{B}_\varepsilon(x)$ and produces several examples. Among those, it selects a point with the maximum loss.

**Zero-order (finite-difference) PGD.** The zero-order PGD is different from the standard PGD in that it uses a finite-difference approximation for $\nabla_{\mathbf{x}} L(h_\theta(x), \hat{y})$ instead of actual gradients. To simplify the presentation, the description below denotes $g(x) = L(h_\theta(x), \hat{y})$. The classic two-sided finite difference method for estimating gradients of the function $g(x)$ uses the following equation:

$$\nabla_{\mathbf{x}} g(\boldsymbol{x})_i \approx \frac{g(\boldsymbol{x}_0, \boldsymbol{x}_1, \ldots, \boldsymbol{x}_i + \alpha, \ldots, \boldsymbol{x}_d) - g(\boldsymbol{x}_0, \boldsymbol{x}_1, \ldots, \boldsymbol{x}_i - \alpha, \ldots, \boldsymbol{x}_d)}{2 * \alpha}. \tag{6}$$

However, this computation can be quite expensive as it requires $O(d)$ inferences, where $d$ is the number of pixels in the image. To reduce complexity, we estimate gradients using block-wise differences.

First we organize all variables into blocks. Then, instead of changing a single variable at a time, we add the same $\alpha$ to all variables in a block and additionally divide the estimate in Eq. (6) by the number of variables in a block. This basic procedure works well for low-dimensional CIFAR images, but not for ImageNet. For ImageNet, we first divide an image into equal-size squares. Then, we combine squares into bigger blocks.

**AutoAttack.** The AutoAttack package (Croce & Hein, 2020a)—or simply AutoAttack—is a suite of attacks that has several variants of a gradient (PGD) attack as well as the query-efficient black-box SQUARE attack (Andriushchenko et al., 2020). Note that both the SQUARE attack and the zero-order PGD attacks are query-based attacks. One of the key contributions of Croce & Hein (2020b) was an Automatic Parameter-free PGD attack with two different loss functions, which are abbreviated as APGD-CE and APGD-DLR.

In addition to this, APGD can be combined with an Expectation Of Transformation (EOT) algorithm (Athalye et al., 2018), which is used to approximate gradients of randomized models. EOT was shown to be effective against randomized defenses (Croce & Hein, 2020a). The third algorithm is a targeted FAB attack, which was shown to be effective against gradient masking (Croce & Hein, 2020b).

The `standard` version of AutoAttack includes all gradient attacks *without* EOT and the SQUARE attack. The `rand` version of AutoAttack has only APGD with EOT where the number of restarts is set to one, but the number of gradient estimation iterations is set to 20.

**Transfer attack.** A transfer attack is an adversarial black-box attack where an example created to fool one (source) model is applied to a different (i.e., target) model. This exploits the phenomenon that adversarial examples often transfer across models, even when they differ in architecture, training data, or parameters (Szegedy et al., 2014).

**Attacks that bypass the front end.** To circumvent the gradient-masking front end, we use two similar "removal" approaches. They both exploit the fact that a front end network is close to the identity transformation, i.e., $f_{\theta_1}(x) \approx x$.

The first approach is a variant of the Backward Pass Differentiable Approximation (BPDA) approach with a straight-through gradient estimator (Athalye et al., 2018). We use a previously described PGD attack and only change the way we compute gradients of the front-end enhanced network $h_{\theta_2}(f_{\theta_1}(x))$. In a general case of a straight-through gradient estimator, one would use $\nabla_{\mathbf{z}} h_{\theta_2}(z)|_{z=f_{\theta_1}(x)}$ instead of $\nabla_{\mathbf{x}} h_{\theta_2}(f_{\theta_1}(x))$. However, due to the front end $f_{\theta_1}(x)$ being approximately equal to the identity transformation, we use $\nabla_{\mathbf{x}} h_{\theta_2}(x)$ directly.

Another approach is a variant of a transfer attack that simply generates the adversarial example $\hat{x}$ for the network $h_{\theta_2}(x)$ and input $x$. Then, the adversarial example $\hat{x}$ is passed to a full composite network $h_{\theta_2}(f_{\theta_1}(x))$.[4] To generate an adversarial example, we use the (gradient) APGD attack from AutoAttack (Croce & Hein, 2020b).

---

[4] Once again, we exploit the fact that $f_{\theta_1}(x) \approx x$.

## 5 Experiments

### 5.1 Experimental Setup

**Evaluation Protocol.** We use three datasets: CIFAR10, CIFAR100 (Krizhevsky et al., 2009), and ImageNet (Deng et al., 2009) all of which have RobustBench leaderboards for $L^\infty$ attacks (Croce et al., 2021). CIFAR10/100 images have a small resolution of 32 by 32 pixels, while ImageNet models use the resolution $224 \times 224$. We use attack radii that are the same as in RobustBench, namely, $\varepsilon = 8/255$ for CIFAR10/CIFAR100 and $\varepsilon = 4/255$ for ImageNet.

All our datasets have large test sets. However, running AutoAttack on the full test sets is quite expensive. It is also unnecessary since we do not aim to establish a true AutoAttack SOTA. Thus, we use randomly selected images: 500 for CIFAR10/CIFAR100 and 200 for ImageNet. The uncertainty in accuracy evaluation due to this sampling is estimated using 99% confidence intervals. Note that in some cases an attack is perfectly successful and every image is classified incorrectly. In such cases, the sample variance is zero and the confidence interval cannot be computed.

**Models.** For our main experiments we use four types of models that can be classified along two dimensions:

- Presence of the front end;
- Single-model vs a three-model randomized ensemble.

A detailed description of these models and their training procedures is given in Section 4.1. Here we only note that four backbone types are used for all three datasets. The only exception is a WideResNet, which is used only for CIFAR10 and CIFAR100.

**Attacks.** We used a combination of attacks described in Section 4.2:

- Standard 50-step PGD with 5 restarts;
- Zero-order (finite-difference) 10-step PGD with 1 restart. For ImageNet, pixels are first grouped into 8x8 squares. For CIFAR datasets, which have very low resolution, the square size is one pixel (no such grouping occurs). Furthermore, these blocks are randomly grouped (5 squares in a group).
- Various combinations of attacks from AutoAttack (Croce & Hein, 2020a). Unless specified otherwise:
  - When attacking a single (i.e., a non-randomized model) we used APGD attacks without EOT (Athalye et al., 2018). Together with the SQUARE attack (Andriushchenko et al., 2020) and FAB-T (Croce & Hein, 2020b) this combination is a `standard` AutoAttack.
  - To attack a randomized ensemble, we use an *expanded* `rand` variant of AutoAttack. AutoAttack `rand` has only APGD attacks with EOT (denoted as APGD/EOT). However, we also use the SQUARE attack.
- BPDA PGD with 10 steps and 5 restarts.
- Transfer attack using AutoAttack APGD.

### 5.2 Discussion of Results

**Summary of results** In this section we briefly review results showing that models with differentiable front ends can, indeed, appear much more robust than they actually are. In particular, when such models are assembled into a randomized ensemble, they appear to achieve near-SOTA robust accuracy virtually without degradation on clean test sets. We will then try to answer the following research questions:

- Is this behavior due to gradient masking?

- Is AutoAttack in principle effective against randomized ensembles?

Table 1: Accuracy for unprotected/standard-training models with a front end (see table notes).

| **Single Model** | | | | | | |
|---|---|---|---|---|---|---|
| Model | No Attack | PGD (standard) | APGD+FAB-T (AutoAttack) | SQUARE (AutoAttack) | PGD (0-order) | PGD (BPDA) |
| CIFAR10 | | | | | | |
| ResNet-18 | 91±3% | 0% | | 1±0% | 3±2% | |
| ResNet-18 +front | 91±3% | 88±4% | 75±5% | 1±1% | 5±2% | 0% |
| VIT-base-16x16 | **97**±2% | 0±1% | | 1±1% | **10**±4% | |
| VIT-base-16x16 +front | **97**±2% | **96**±2% | **85**±4% | 1±2% | **10**±3% | 0±1% |
| CIFAR100 | | | | | | |
| ResNet-18 | 68±6% | 0±1% | | 1±1% | 4±3% | |
| ResNet-18 +front | 68±5% | 65±5% | 48±6% | 2±1% | 4±2% | 0±1% |
| VIT-base-16x16 | **84**±5% | 1±1% | | 2±2% | 8±3% | |
| VIT-base-16x16 +front | **84**±5% | **80**±5% | **59**±6% | 3±2% | 8±4% | 1±1% |
| ImageNet | | | | | | |
| ResNet-18 | 69±9% | 0% | | 2±2% | 23±8% | |
| ResNet-18 +front | 70±8% | 68±9% | 28±8% | 2±3% | 23±8% | 0% |
| VIT-base-16x16 | **83**±7% | 0% | | 18±8% | 60±9% | |
| VIT-base-16x16 +front | 82±8% | **79**±8% | **48**±9% | **21**±8% | **62**±9% | 0% |
| **Randomized Ensemble** | | | | | | |
| Model | No Attack | PGD (standard) | APGD/EOT+SQUARE (AutoAttack) | SQUARE (AutoAttack) | PGD (0-order) | PGD (BPDA) |
| CIFAR10 | | | | | | |
| ResNet-18 3× | 92±3% | 2±2% | | 76±5% | 66±6% | |
| ResNet-18 +front 3× | 91±4% | 89±4% | 70±6% | 74±5% | 64±5% | 3±2% |
| VIT-base-16x16 3× | **97**±2% | 1±1% | | 78±5% | 64±6% | |
| VIT-base-16x16 +front 3× | 96±2% | **96**±2% | **74**±5% | 81±4% | 65±6% | 1±1% |
| CIFAR100 | | | | | | |
| ResNet-18 3× | 69±5% | 5±3% | | 51±6% | 49±5% | |
| ResNet-18 +front 3× | 66±6% | 67±5% | 40±6% | 54±6% | 46±6% | 8±3% |
| VIT-base-16x16 3× | 85±4% | 2±2% | | **64**±6% | 46±5% | |
| VIT-base-16x16 +front 3× | **86**±4% | **81**±5% | **54**±6% | 63±5% | 45±6% | 3±1% |
| ImageNet | | | | | | |
| ResNet-18 3× | 69±9% | 0% | | 2±2% | 23±8% | |
| ResNet-18 +front 3× | 70±8% | 68±9% | 42±10% | 48±9% | 33±9% | 0±2% |
| VIT-base-16x16 3× | **83**±7% | 0% | | 18±7% | 58±10% | |
| VIT-base-16x16 +front 3× | 82±8% | **80**±7% | **63**±9% | **69**±9% | **68**±9% | 0% |

Test sample has 500 images for CIFAR10/CIFAR100 and 200 images for ImageNet:
We show 99% confidence intervals (CI) when applicable: Note that CIs can not be computed for attacks with 100% success rate.
APGD/EOT+SQUARE is a combination of APGD with EOT (as in `rand` version of AutoAttack) followed by SQUARE attack.
We do not run APGD or BPDA attacks when the accuracy ≈ 0% under the *standard* PGD attack.

Table 2: Summary of results for unprotected/standard-training models with a front end (see table notes).

| Dataset | $\varepsilon$-attack | Single model (stand. train.) | Best Randomized Ensembles (adv. train. with frozen backbones) | | | Best RobustBench (June 2025) original dataset | | extra data | |
|---|---|---|---|---|---|---|---|---|---|
| | | No Attack | No Attack | AutoAttack (APGD/EOT+SQUARE) | PGD (BPDA) | No Attack | AutoAttack (standard) | No Attack | AutoAttack (standard) |
| CIFAR10 | 8/255 | 95±3% | 96±2% | 74±5% | 5±2% | 92.2% | 66.6% | 93.7% | 73.7% |
| CIFAR100 | 8/255 | 84±5% | 86±4% | 54±7% | 3±1% | 69.2% | 36.9% | 75.2% | 42.7% |
| ImageNet | 4/255 | 85±4% | 82±8% | 63±9% | 0% | 78.9% | 59.6% | | |

Attack $\varepsilon = 8/255$. Test sample has 500 images for CIFAR10/CIFAR100 and 200 images for ImageNet: We show 99% confidence intervals (CI) when applicable: Note that CIs can not be computed for attacks with 100% success rate.

Table 3: Accuracy for adversarially-trained models with a front end on CIFAR10 (see table notes).

| | | | Single model | | | |
|---|---|---|---|---|---|---|
| $\varepsilon$-train | No Attack | +APGD-CE | +APGD-T | +FAB-T | +SQUARE | Transfer (AutoAttack) |
| 2/255 | 95.2% | 92.6% | 92.4% | 92.4% | 41.0±4.3% | 15.2±3.2% |
| 4/255 | 92.2% | 90.8% | 90.6% | 90.6% | 56.4±4.4% | 32.6±4.1% |
| 8/255 | 84.8% | 83.8% | 82.6% | 82.6% | 60.8±4.3% | 43.6±4.4% |
| | | | Randomized Ensemble | | | |
| $\varepsilon$-train | No Attack | +APGD-CE | +APGD-DLR/EOT | | +SQUARE | Transfer (AutoAttack) |
| 2/255 | 94.4% | 93.8% | 92.4% | | 90.8±2.5% | 18.2±3.6% |
| 4/255 | 92.2% | 91.4% | 90.6% | | 89.0±2.8% | 35.6±4.3% |
| 8/255 | 84.8% | 83.2% | 82.2% | | 80.2±3.5% | 45.2±4.4% |

Attack $\varepsilon = 8/255$. Test sample has 500 images: We show 99% confidence intervals (CI).
To simplify exposition we do not post CIs for intermediate results, but only for the final attacks.

In our experiments we focused primarily on combining standard models with a front end (except for CIFAR10). This was done to reduce experimentation cost. Indeed, training adversarially robust models for ImageNet is computationally intensive while high-accuracy models trained using the standard objective and original/clean data are readily available.

The key outcomes of these experiments are presented in Tables 1 and 2. Table 1 includes results only for ResNet-18 (He et al., 2016) and ViT (Dosovitskiy et al., 2021). Expanded variants of Table 1, which include additional models are provided in the Appendix A.3 (Tables 10, 11, 12).

Using ResNet18 on CIFAR-10, we experimented with removing the skip connection from the DnCNN front-end. By design, the skip connection allows the convolutional layers to focus on learning how to suppress corruption of small magnitude, effectively acting as a near-identity transformation. Without the skip connection, training makes virtually no progress even after dozens of epochs. Interestingly, increasing the learning rate in this case leads to behavior similar to that of standard adversarial training.

From Table 2 it can be seen that randomized ensembles, which include front-end enhanced models, rival the best models on RobustBench (Croce et al., 2021), even when RobustBench models were trained using additional data. At the same time, we do not observe degradation on clean data, which indicates that a front-end should largely preserve original images. However, all front-end enhanced models have nearly zero accuracy under the BPDA PGD attack. At the same time, neither the gradient attacks nor the SQUARE (Andriushchenko et al., 2020) attack are particularly effective.

The SQUARE attack is quite effective against a single model without a front end. On CIFAR10 and CIFAR100 datasets (see Table 1 and Tables 10, 11 in Appendix A.3) it degrades accuracy of all models to nearly zero. For ImageNet and ViT, the accuracy under attack is also quite low (about 20%). However, it is much less successful against randomized ensembles, even when individual models are without front ends. We also find that in some cases the zero-order PGD can be more successful than other attacks. However, the attack success rate is still well below the expected 100%. To reiterate, both zero-order PGD and the SQUARE attack are particularly ineffective against randomized ensembles with front-end enhanced models.

Table 4: Accuracy under attack for adversarially-trained models on CIFAR10 for standard vs zero-Order PGD (see table notes).

|  | No Attack | PGD (0-order) | PGD (standard) |
|---|---|---|---|
| ResNet-18 | 65±2% | 53±2% | 42±2% |
| ResNet-50 | 76±2% | 62±2% | 53±2% |
| ResNet-101 | 78±2% | 65±2% | 52±2% |

All models are trained using a standard adversarial training procedure without a front end (training $\varepsilon = 8/255$).
Test sample has 2000 images: 99% confidence intervals are shown.

What happens when an adversarially trained model is combined with a front end? We carried out a detailed investigation using a sample of CIFAR10 images and presented results in Table 3. The main difference here is that we report accuracy after each attack from AutoAttack (Croce & Hein, 2020a). Thus, we can see how successful each type of attack is.

From Table 3 we can see that adversarial training even with small $\varepsilon = 2/255$, makes the model nearly "impervious" to gradient attacks when such a model is combined with a front-end. For example, a combination of all gradient attacks degrades the accuracy of a single front-end "protected" model from 95.2% to only 92.4%. At the same time, the true robust accuracy of this model does not exceed 15.2% (we know that the accuracy under the transfer attacks is about 15%, but it might be possible to further reduce it using some unknown approach). In contrast, the SQUARE attack could reduce accuracy to only 41%. In the case of a randomized ensemble, though, it is not effective and the combination of attacks can reduce accuracy to only 90.8%. Curiously enough, adversarial training seems to increase resistance to both gradient and black-box attacks.

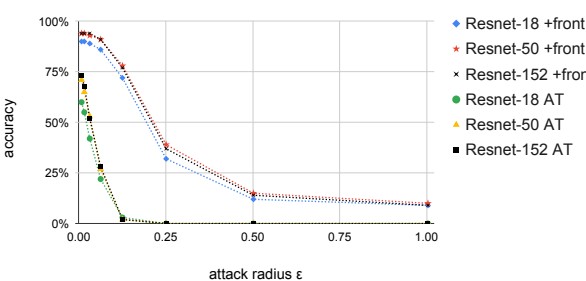

Figure 2: CIFAR10: Accuracy under attack for different attack radii for models with a front-end and models that underwent traditional (multi-epoch) adversarial training without using a front-end (denoted with AT). Accuracy of front-end equipped models is decreasing much slower than the accuracy of AT models, which is indicative of gradient masking.

**Is it gradient masking?** It should now be clear that AutoAttack (and/or its individual attacks) can greatly exaggerate performance of both single models and (especially) randomized ensembles. There are several indicators that this is primarily due to gradient masking. First, if we plot accuracy of the models for attacks of varying radius $\varepsilon$ (as recommended by Tramèr et al. 2020), we can see that the accuracy of adversarially-trained models without front ends quickly decreases as the attack radius $\varepsilon$ grows. This happens for at least two reasons:

- Since the model is trained to resist perturbations of a radius at most $\varepsilon = 8/255$, it is easily "fooled" when the attack radius increases;

- Large perturbations can alter the underlying class semantics, so we should not expect the model to predict the initial class as the radius grows.

In contrast, the accuracy of the front-end equipped models declines noticeably slower and does not reach zero when $\varepsilon = 1$, which is the maximum possible distortion (remember that pixel intensity ranges from 0 to 1).

Second, we can see that black-box attacks, namely SQUARE and zero-order PGD, can be substantially more effective than standard gradient attacks. In that, for a typical adversarially trained model, the zero-order PGD is less effective (see Table 4). This suggests that the gradients provide little useful information about the true optimization direction—a characteristic phenomenon of gradient masking.

Third, we can see that front-end bypassing attacks—such as BPDA PGD—make front-end "protected" models much less accurate. The ineffectiveness of standard gradient-based attacks, contrasted with the success of stronger approaches such as transfer attacks, is a well-recognized indicator of gradient masking Tramèr et al. (2020).

Although gradient masking is a known phenomenon, it is unusual to observe in fully differentiable models without gradient-diminishing or gradient-shattering components. For example, Tramèr et al. (2020) describe a case when gradient masking was caused by an additional "log-softmaxing" which is known to reduce "gradient flows". However, in our case, a gradient-masking culprit is a front end, which is a fully-convolutional network with a skip connection and batch-normalization layers (see § A.2 where we describe checking for numerical abnormalities). Both of these architectural components were instrumental to training deep neural networks, which otherwise would suffer from vanishing or exploding gradients (Szegedy et al., 2014; He et al., 2016).

Table 6: Accuracy under attack for diverse randomized ensembles of varying adversarial robustness on CIFAR10 (500 images).

| $\varepsilon$-train | PGD (ensemble average) | PGD (full ensemble) | AutoAttack |
|---|---|---|---|
| 2/255 | 30±2% | 64±6% | 29±5% |
| 4/255 | 47±2% | 67±5% | 42±6% |
| 8/255 | 52±3% | 64±6% | 47±6% |

**Is AutoAttack in principle effective against randomized ensembles?** We observed that AutoAttack can be ineffective against randomized ensembles. In particular, on a sample of CIFAR10 images, we can achieve an accuracy of 90.8±2.5% (99% CI) for a model that has true robust accuracy below 20%. Is AutoAttack, in principle, weak against randomized ensembles or is there something special about our ensemble models?

First of all, we confirmed that even standard PGD can be extremely effective against randomized ensembles where individual models underwent only standard training using clean data. Indeed, from Table 1 and Tables 10, 11 in Appendix A.3 we see that the accuracy of such ensembles is nearly zero under a standard PGD attack. This was an unexpected finding because such attacks do not use averaging techniques such as EOT and each gradient computation uses a mix of models with different random weights.

A more careful investigation showed that—despite the lack of EOT—because all models in the ensemble share the same architecture, the standard PGD finds adversarial examples that work for *all* models in the ensemble. To prevent this, we trained ensembles using three different types of backbone models, namely, ResNet-152 (He et al., 2016), wide ResNet-28-10 (Zagoruyko & Komodakis, 2016), and ViT (Dosovitskiy et al., 2021). We did not use any front ends and trained three ensembles using three values of attack $\varepsilon$: 2/255, 4/255, and 8/255. For each model in the ensemble we computed the accuracy under a standard PGD attack (with $\varepsilon = 8/255$) and then obtained ensemble averages, which ranged from 30% to 52% depending on the training-time $\varepsilon$.

As expected, the smaller $\varepsilon$ was during training, the lower was the robust accuracy of the models. However, when we attacked a randomized ensemble using a 50-step PGD with five restarts, we achieved the accuracy close to 65% (see Table 6). Thus, the randomized ensemble of diverse models can fool the standard PGD attack without EOT. In contrast, the accuracy under AutoAttack was slightly lower (as expected) than ensemble-average robust accuracy (see Table 6), thus, confirming that AutoAttack can indeed be very effective against randomized ensembles.

**Can randomized ensembling offer a practical defense?** While our primary focus is not on designing or benchmarking a specific randomized-ensemble scheme, it is worth discussing whether such an approach could meaningfully boost robustness. To this end, we should first consider plausible attack scenarios. Sitawarin

Table 5: Calculating the lower bound for the accuracy of the randomized ensemble under a transfer attack with a single source model.

We calculate a lower bound for the expected accuracy of the randomized ensemble under a transfer attack, assuming the following:

- An attacker creates an attack by using one model at a time, i.e., there is no ensembling of source models;

- A source model is one of the models used in the randomized ensemble;

- Individual models in the randomized ensemble are selected with equal probabilities.

According to Table 8 (in § A.1):

- $A(i \to i) \approx 0$ (self-attack);

- $A_{k \to i} \geq 44.2\%$ if both $i$ and $k$ are different ResNet models (attacking a ResNet using a different ResNet model);

- $A_{k \to i} \geq 73.6\%$ if $i$ is a ViT model and $k$ is a ResNet model (attacking a ViT model using a ResNet model);

- $A_{k \to i} \geq 65.4\%$ if $i$ is a ResNet model and $k$ is a ViT model (attacking a ResNet model using a ViT model);

- $A_{k \to i} \geq 55.9\%$ if both $i$ and $k$ are ViT models (attacking a ViT model using a different ViT model).

First, there is a 10% chance that the source model and the target model are the same. In this case, the average accuracy is near zero. For a source ResNet model, in four cases out of 10, $A_{k \to i} \geq 44.2\%$ (when a target model is also a ResNet model) and in five cases out of ten $A_{k \to i} \geq 73.6\%$ (when the target is a ViT). Likewise for a source ViT model, in four cases out of ten $A_{k \to i} \geq 55.9\%$ and in five cases out of ten $A_{k \to i} \geq 65.4\%$. Therefore, for a source ResNet model, the lower bound for the expected accuracy is:

$$0\% \cdot 0.1 + 44.2\% \cdot 0.4 + 73.6\% \cdot 0.5 \approx 54.6\%.$$

For a source ViT model, the lower bound for the expected accuracy is:

$$0\% \cdot 0.1 + 55.9\% \cdot 0.4 + 65.4\% \cdot 0.5 \approx 55\%$$

Table 7: Calculating the expected accuracy of the randomized ensemble under the transfer attack that uses the all-model ensemble.

---

We calculate the expected accuracy of the randomized ensemble under attack, assuming the following:

- The attacker creates an attack using an ensemble model, which includes *all* models;

- Individual models in the randomized ensemble are selected with equal probabilities.

According to Table 8 (in § A.1):

- $A_{\text{ensemble} \to i} = 18.1\%$ if $i$ is a ResNet model;

- $A_{\text{ensemble} \to i} = 26.1\%$ if $i$ is a ViT model.

Because a ViT model is randomly selected 50% of the time (in the ensemble) the expected accuracy is equal to:

$$18.1\% \cdot 0.5 + 26.1\% \cdot 0.5 \approx 22\%$$

---

et al. (2024); Qin et al. (2021) showed that randomization thwarts several popular query-based attacks (which we also confirm for the AutoAttack case). Therefore, a typical attacker has to rely on transfer attacks or on hybrid approaches that combine transfer attacks with query-based attacks.

Assume that the defender has $n$ models and makes a prediction by randomly selecting a model according to probabilities $\{p_i\}$. The attacker has $m$ models from which they select a source model $k$ to generate an adversarial example. Given such a setup, the expected accuracy of the randomized ensemble is:

$$A(k) = \sum_{i=1}^{n} A_{k \to i} \cdot p_i, \tag{7}$$

where $A_{k \to i}$ is the expected accuracy of model $i$ when making a prediction on an adversarial example transferred from model $k$. The best the attacker can do in this situation is to select $k = \arg\min_{k=1}^{m} A(k)$ that minimizes the expected accuracy under attack. In the worst case for the defender, the attacker has detailed knowledge about model architectures and parameters and, thus, be able to compute $A(k)$ even without querying the randomized ensemble. Nonetheless, even in the absence of such knowledge, the attacker may be able to estimate $A(k)$ by probing the ensemble for different values of $k$—albeit this may not always be practical due to a significant reduction in efficiency (a reliable estimate requires a lot of queries for each $k$).

To make things concrete, let us consider 10 models used in our transferability study for ImageNet (see § A.1 and Table 8). This study includes five ResNet and five ViT models trained using clean data. Because these models were not trained adversarially, even small-radius PGD attacks "drive" their accuracy to zero, i.e., $A(i \to i) \approx 0$. However, $A(k \to i)$ is well above zero for $i \neq k$.

To simplify the presentation, we moved the calculations of the expected accuracy under attack to Table 5. In summary, no matter what the source model is, the defender is guaranteed to have the accuracy under attack above 54%. This is about 23 percentage point decrease compared to average accuracy on the clean data, which is 77%. However, it is still markedly better than a nearly 0% accuracy under attack, and importantly, the clean data accuracy is fully retained.

Instead of using individual models, the attacker can use ensembles as source models, which are known to produce stronger attacks (Liu et al., 2017; Gubri et al., 2022). A particularly unfortunate for the defender situation is when the attacker can ensemble *all* models to generate an attack. In this case (see Table 7 for details), the accuracy under attack is only 22%, which is a 2.5× decrease compared to the case when the attacks are generated using a single model.

It should now be clear that randomization can be a practical defense only if one combines models with low mutual attack transferability. Further exploration of the feasibility of this approach is out of the scope of this work. However, we would like to mention several approaches to achieve this objective. First, Bai et al. (2024) found there is only limited transferability among models trained using adversarial attacks with different strengths, which is also confirmed by our own experiments (see Table 9 in § A.1). Not only transfer is limited when the source model is trained on clean-data, but, surprisingly, adversarial examples coming from more robust models do not work well for less robust models. Second—as discussed in § 2—there are additional training approaches that reduce mutual transferability among models (Sharif et al., 2019; Bui et al., 2021; Sitawarin et al., 2024). Third, one can modify existing architectures in controlled ways, e.g., by changing normalization approaches (Dong et al., 2022) or precision (Fu et al., 2021). This can be used to further improve the architectural diversity of the randomized ensemble.

## 6 Conclusion

We discovered that training a fully differentiable front end, which also has a skip connection, with a frozen classification model can result in a model that is unusually resistant to gradient attacks while retaining accuracy on original data. We provided evidence that this behavior is due to gradient masking. Although the gradient masking phenomenon is not new, the degree of this masking is remarkable for fully differentiable models lacking gradient-shattering components like JPEG compression or other elements known to diminish gradients. The objective of this paper is to document this phenomenon and expose weaknesses of AutoAttack (Croce & Hein, 2020a) rather than to establish new SOTA in adversarial robustness. We conclude the paper with a discussion of whether randomized ensembling can function as a practical defense.

## 7 Acknowledgments

This work was sponsored by DARPA grant HR11002020006.

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

## A  Appendix

### A.1  Transfer Attack Study

We have carried out a study of the effectiveness of transfer attacks using two different scenarios, in particular, to assess feasibility of using randomized ensembles as practical defenses. Attacks were generated using a 50-step PGD: $\varepsilon = 8/255$ for CIFAR datasets and $\varepsilon = 4/255$ for ImageNet. We used the same subsets of CIFAR10, CIFAR100, and ImageNet as in the main experiments.

In the first scenario, we assessed transferability among ten different architectures: Five of the models belong to the ResNet class (from ResNet-18 to ResNet-152) (He et al., 2016) and five to the ViT (Dosovitskiy et al., 2021) class. ViT models differed in their sizes (tiny, small, base) and input patch sizes. In addition to measuring "clean" accuracy and accuracy under a PGD attack, we computed the accuracy under a transfer attack for several source-model architectures. A source model can be either a single model or an ensemble, but a target model is always one of the above-mentioned ResNet or ViT models. The source-models included:

- The same model as the target model (the case of a self-attack).

- An individual model whose architecture is different from the target model. To simplify presentation, we only show accuracy for source models producing the strongest attack (i.e., that led to the lowest accuracy). In doing so, we show results for the best ResNet and the best ViT model separately.

- An ensemble of models. We tested two types of ensembles: (1) a *leave-one-out* ensemble that does not include the target model, and (2) an all-inclusive ensemble that has all models of a given type (or types). All ensembles were additionally divided into three groups:
  - An ensemble of ResNet models, which has either all ResNet models (an inclusive ensemble) or all ResNet models except the target model (a leave-one-out ensemble);
  - An ensemble of ViT models, which has either all ViT models (an inclusive ensemble) or all ViT models except the target model (a leave-one-out ensemble);
  - An all-models ensemble, which has either all 10 models (an inclusive ensemble) or all models except the target one (a leave-one-out ensemble);

To simplify presentation, we only show summary results obtained by averaging accuracies for all target models of the same class. From Table 8, we can see that the accuracy under the self-attack is always under 5%. This is expected because models were not trained adversarially. However, attacks from different architectures are less successful, especially attacks from ResNet to ViT models and vice versa. Moreover, generating attacks using ensembles produces stronger attacks, which is well known from the literature (Liu et al., 2017; Gubri et al., 2022). Finally, we note that on ImageNet we observe the lowest transferability of attacks.

In our next experiment, we tested five ResNet and two ViT architectures on CIFAR10. Each architecture variant was trained using a different scenario. Scenarios include (1) training on clean data and (2) adversarial training with different attack strengths: from $\varepsilon = 2/255$ to $8/255$. The attack radius was $\varepsilon = 8/255$.

From Table 9, we can see that transferring an attack from a model trained on clean data to a model trained with the attack radius $\varepsilon = 2/255$ reduces the accuracy by only 6.7 percentage points. This is totally unsurprising since the clean-data model is not supposed to produce strong attacks for robust models. However, transferring attacks in the other direction—i.e., from the model trained adversarially with $\varepsilon = 2/255$ to a model trained on clean data—yields 34.7% accuracy, still massively higher than the 0.4% achieved under the self-attack.

### A.2  Gradient Tracing: Checking for Numerical Instabilities

To make sure our results were not affected by numerical instabilities (or by exploding and vanishing gradients), we "traced" gradients for ResNet-152 (He et al., 2016) backbone on all three datasets. We used PGD

Table 8: Transfer attack accuracy among different model architectures (see table notes). We show averages for ResNet and ViT models.

| source \ target | CIFAR10 | | CIFAR100 | | ImageNet | |
|---|---|---|---|---|---|---|
| | ResNet | ViT | ResNet | ViT | ResNet | ViT |
| *Source model and target models are different* | | | | | | |
| No attack | 92.7% | 96.5% | 73.0% | 80.3% | 74.7% | 79.1% |
| Self-attack | 0.3% (-92%) | 0.6% (-96%) | 2.2% (-71%) | 1.4% (-79%) | 0.0% (-75%) | 3.1% (-76%) |
| Best ResNet | 33.2% (-60%) | 80.3% (-16%) | 31.5% (-41%) | 59.9% (-20%) | 44.2% (-31%) | 73.6% (-5%) |
| Best ViT | 67.0% (-26%) | 12.9% (-84%) | 50.4% (-23%) | 16.9% (-63%) | 65.4% (-9%) | 55.9% (-23%) |
| Leave-One-Out (LOT) ensembles: *target models are excluded* | | | | | | |
| ResNet ensemble | 24.3% (-68%) | 63.8% (-33%) | 19.6% (-53%) | 46.6% (-34%) | 32.9% (-42%) | 71.1% (-8%) |
| ViT ensemble | 53.9% (-39%) | 10.0% (-87%) | 39.2% (-34%) | 12.3% (-68%) | 60.6% (-14%) | 55.2% (-24%) |
| All ensemble | 25.6% (-67%) | 39.0% (-58%) | 20.9% (-52%) | 23.2% (-57%) | 34.9% (-40%) | 59.1% (-20%) |
| *All-inclusive* ensembles: All models of a given type are included | | | | | | |
| ResNet ensemble | 15.4% (-77%) | 62.8% (-34%) | 11.4% (-62%) | 46.9% (-33%) | 13.8% (-61%) | 71.0% (-8%) |
| ViT ensemble | 52.9% (-40%) | 6.4% (-90%) | 38.8% (-34%) | 6.8% (-73%) | 60.5% (-14%) | 15.4% (-64%) |
| All ensemble | 17.8% (-75%) | 35.3% (-61%) | 13.9% (-59%) | 17.8% (-63%) | 18.1% (-57%) | 26.1% (-53%) |

We experiment with five ResNet models and five ViT models: Best ResNet and Best ViT denote the most effective attack using ResNet or ViT.
Model are trained using "clean" data, i.e., without adversarial attacks.
In round brackets we show the loss of accuracy compared to "clean" accuracy (no attack), which we round to a nearest integer.

Table 9: CIFAR10: average transfer attack accuracy among models with the same architecture, but trained with different attack strengths: In round brackets we show the loss of accuracy compared to "clean" accuracy (no attack). Diagonal row shows the self-attack accuracy (attack is generated using the same model). We tested five ResNet and two VIT architectures.

| source \ target ($\varepsilon$-train) | No attack | Accuracy under transfer attack | | | |
|---|---|---|---|---|---|
| | | no adv. train. | 2/255 | 4/255 | 8/255 |
| no adv. train | 93.7% | 0.4% (-93.4%) | 83.1% (-6.7%) | 83.1% (-3.8%) | 75.3% (-2.6%) |
| 2/255 | 89.8% | 34.7% (-59.0%) | 34.6% (-55.3%) | 60.9% (-26.0%) | 65.9% (-12.0%) |
| 4/255 | 86.9% | 52.2% (-41.5%) | 51.8% (-38.0%) | 46.9% (-40.0%) | 60.4% (-17.5%) |
| 8/255 | 77.9% | 72.5% (-21.2%) | 67.1% (-22.7%) | 61.7% (-25.2%) | 51.8% (-26.1%) |

with 50 steps and a single restart. We did this for all datasets using a sample of 500 images. We compared gradients between two scenarios:

- Attacking a model trained on clean data;

- Attacking a front-end "enhanced" model that was adversarially trained.

We found no apparent anomalies. In particular:

- No gradient was ever NaN or $\pm\infty$;

- In both cases only a modest fraction (10-30%) of absolute gradient values were small ($< 0.01$);

- In both cases, mean and median gradients (per image) tended to be close to zero;

- 99% of gradients absolute values were under 2000;

- In that, the distribution of gradient values for front-end "enhanced" models tended to be more skewed with more extreme values in lower or higher quantiles (by 1-2 order of magnitudes).

### A.3 Additional Experimental Results

Table 10: Detailed results for CIFAR10 (see table notes).

| Single Model | | | | | | |
|---|---|---|---|---|---|---|
| Model | No Attack | PGD (standard) | APGD+FAB-T (AutoAttack) | SQUARE (AutoAttack) | PGD (0-order) | PGD (BPDA) |
| ResNet-18 | 91±3% | 0% | | 1±0% | 3±2% | |
| ResNet-18 +front | 91±3% | 88±4% | 75±5% | 1±1% | 5±2% | 0% |
| ResNet-50 | 95±3% | 0% | | 0±1% | 4±3% | |
| ResNet-50 +front | 95±3% | 93±3% | 78±5% | **2**±2% | 5±3% | 0±1% |
| ResNet-152 | 94±3% | 0% | | 1±2% | 6±3% | |
| ResNet-152 +front | 94±3% | 92±3% | 81±4% | **2**±2% | 7±3% | 0% |
| WideResNet-28-10 | 96±2% | 0% | | 0% | 8±3% | |
| WideResNet-28-10 +front | 95±3% | 91±3% | 65±5% | 1±1% | **10**±4% | 1±1% |
| VIT-base-16x16 | **97**±2% | 0±1% | | 1±1% | **10**±4% | |
| VIT-base-16x16 +front | **97**±2% | **96**±2% | **85**±4% | 1±2% | **10**±3% | 0±1% |
| Randomized Ensemble | | | | | | |
| Model | No Attack | PGD (standard) | APGD/EOT+SQUARE (AutoAttack) | SQUARE (AutoAttack) | PGD (0-order) | PGD (BPDA) |
| ResNet-18 3× | 92±3% | 2±2% | | 76±5% | 66±6% | |
| ResNet-18 +front 3× | 91±4% | 89±4% | 70±6% | 74±5% | 64±5% | 3±2% |
| ResNet-50 3× | 95±3% | 3±2% | | 80±4% | 69±5% | |
| ResNet-50 +front 3× | 96±2% | 94±3% | 71±5% | 81±4% | 69±5% | 5±2% |
| ResNet-152 3× | 94±3% | 3±2% | | **84**±4% | 70±5% | |
| ResNet-152 +front 3× | 95±2% | 94±3% | 71±5% | 79±5% | **74**±5% | **6**±3% |
| WideResNet-28-10 3× | 96±2% | 3±2% | | 77±5% | 45±6% | |
| WideResNet-28-10 +front 3× | 95±3% | 92±3% | 60±6% | 79±4% | 51±5% | 3±2% |
| VIT-base-16x16 3× | **97**±2% | 1±1% | | 78±5% | 64±6% | |
| VIT-base-16x16 +front 3× | 96±2% | **96**±2% | **74**±5% | 81±4% | 65±6% | 1±1% |

Test sample has 500 images: 99% confidence intervals are shown. Each random ensemble has 3 models.

APGD/EOT+SQUARE is a combination of APGD with EOT (as in `rand` version of AutoAttack) followed by SQUARE attack.

Best outcomes are shown in bold. Best results are computed **separately** for single models and ensembles.

We show 99% confidence intervals: They cannot be computed when the attack has 100% success rate.

We do not run additional PGD attacks (except 0-order PGD) when the accuracy ≈ 0% with the standard PGD attack.

Table 11: Detailed results for CIFAR100 (see table notes).

| Single Model | | | | | | |
|---|---|---|---|---|---|---|
| Model | No Attack | PGD (standard) | APGD+FAB-T (AutoAttack) | SQUARE (AutoAttack) | PGD (0-order) | PGD (BPDA) |
| ResNet-18 | 68±6% | 0±1% | | 1±1% | 4±3% | |
| ResNet-18 +front | 68±5% | 65±5% | 48±6% | 2±1% | 4±2% | 0±1% |
| ResNet-50 | 75±5% | 1±0% | | 1±2% | 6±3% | |
| ResNet-50 +front | 77±5% | 70±6% | 53±6% | 2±2% | 7±3% | 1±1% |
| ResNet-152 | 76±5% | 1±2% | | 2±1% | 9±3% | |
| ResNet-152 +front | 74±5% | 71±5% | 51±6% | **5**±2% | **11**±4% | 2±2% |
| WideResNet-28-10 | 76±5% | 2±1% | | 0±1% | 7±3% | |
| WideResNet-28-10 +front | 75±5% | 67±5% | 35±5% | 2±1% | 8±3% | **3**±1% |
| VIT-base-16x16 | **84**±5% | 1±1% | | 2±2% | 8±3% | |
| VIT-base-16x16 +front | **84**±5% | **80**±5% | **59**±6% | 3±2% | 8±4% | 1±1% |
| Randomized Ensemble | | | | | | |
| Model | No Attack | PGD (standard) | APGD/EOT+SQUARE (AutoAttack) | SQUARE (AutoAttack) | PGD (0-order) | PGD (BPDA) |
| ResNet-18 3× | 69±5% | 5±3% | | 51±6% | 49±5% | |
| ResNet-18 +front 3× | 66±6% | 67±5% | 40±6% | 54±6% | 46±6% | 8±3% |
| ResNet-50 3× | 73±5% | 6±2% | | 57±6% | 49±6% | |
| ResNet-50 +front 3× | 75±5% | 72±6% | 46±6% | 58±6% | 51±6% | 9±3% |
| ResNet-152 3× | 76±5% | 10±3% | | 58±6% | 49±6% | |
| ResNet-152 +front 3× | 75±5% | 71±5% | 48±6% | 57±6% | **53**±6% | **10**±4% |
| WideResNet-28-10 3× | 78±5% | 5±2% | | 52±5% | 31±5% | |
| WideResNet-28-10 +front 3× | 76±5% | 70±5% | 34±6% | 52±6% | 33±6% | 5±2% |
| VIT-base-16x16 3× | 85±4% | 2±2% | | **64**±6% | 46±5% | |
| VIT-base-16x16 +front 3× | **86**±4% | **81**±5% | **54**±6% | 63±5% | 45±6% | 3±1% |

Test sample has 500 images: 99% confidence intervals are shown. Each random ensemble has 3 models.

APGD/EOT+SQUARE is a combination of APGD with EOT (as in `rand` version of AutoAttack) followed by SQUARE attack.

Best outcomes are shown in bold. Best results are computed **separately** for single models and ensembles.

We show 99% confidence intervals: They cannot be computed when the attack has 100% success rate.

We do not run additional PGD attacks (except 0-order PGD) when the accuracy $\approx 0\%$ with the standard PGD attack.

Table 12: Detailed results for ImageNet (see table notes).

| Single Model | | | | | | |
|---|---|---|---|---|---|---|
| Model | No Attack | PGD (standard) | APGD+FAB-T (AutoAttack) | SQUARE (AutoAttack) | PGD (0-order) | PGD (BPDA) |
| ResNet-18 | 69±9% | 0% | | 2±2% | 23±8% | |
| ResNet-18 +front | 70±8% | 68±9% | 28±8% | 2±3% | 23±8% | 0% |
| ResNet-50 | 76±8% | 0% | | 7±5% | 40±9% | |
| ResNet-50 +front | 76±7% | 66±9% | 31±9% | 8±5% | 40±10% | 0% |
| ResNet-152 | 78±8% | 0% | | 12±5% | 46±9% | |
| ResNet-152 +front | 78±8% | 74±9% | 26±8% | 16±6% | 47±9% | 0% |
| VIT-base-16x16 | **83**±7% | 0% | | 18±8% | 60±9% | |
| VIT-base-16x16 +front | 82±8% | **79**±8% | **48**±9% | **21**±8% | **62**±9% | 0% |
| **Randomized Ensemble** | | | | | | |
| Model | No Attack | PGD (standard) | APGD/EOT+SQUARE (AutoAttack) | SQUARE (AutoAttack) | PGD (0-order) | PGD (BPDA) |
| ResNet-18 3× | 69±9% | 0% | | 2±2% | 23±8% | |
| ResNet-18 +front 3× | 70±8% | 68±9% | 42±10% | 48±9% | 33±9% | 0±2% |
| ResNet-50 3× | 76±8% | 0% | | 8±4% | 38±8% | |
| ResNet-50 +front 3× | 76±7% | 72±8% | 50±9% | 55±9% | 50±9% | 0% |
| ResNet-152 3× | 78±8% | 0% | | 12±5% | 44±9% | |
| ResNet-152 +front 3× | 78±7% | 74±8% | 48±10% | 53±9% | 53±9% | 0% |
| VIT-base-16x16 3× | **83**±7% | 0% | | 18±7% | 58±10% | |
| VIT-base-16x16 +front 3× | 82±8% | **80**±7% | **63**±9% | **69**±9% | **68**±9% | 0% |

Test sample has 200 images: 99% confidence intervals are shown. Each random ensemble has 3 models.

APGD/EOT+SQUARE is a combination of APGD with EOT (as in `rand` version of AutoAttack) followed by SQUARE attack.

Best outcomes are shown in bold. Best results are computed **separately** for single models and ensembles.

We show 99% confidence intervals: They cannot be computed when the attack has 100% success rate.

We do not run additional PGD attacks (except 0-order PGD) when the accuracy ≈ 0% with the standard PGD attack.

