# OpenReview forum: "A Curious Case of Remarkable Resilience to Gradient Attacks via Fully Convolutional and Differentiable Front End with a Skip Connection"
_TMLR — Accepted by TMLR_

### Review · Reviewer_nhBn · 2024-04-15

**Summary Of Contributions:**

This paper explores front-end enhanced neural models that combine a frozen backbone classifier with a differentiable and convolutional model to improve the adversarial robustness. Specifically, the authors utilize several models for random ensembles with front ends, resulting in models that maintained the accuracy of the backbone classifier while exhibiting significant resistance to gradient attacks. The study demonstrates that this resistance is primarily due to gradient masking. The proposed framework as well as the observations have been validated in multiple datasets and model types. The authors also discuss the limitations of black-box attacks and propose the use of randomized ensembles to counteract them. While the goal of the paper is not to establish SOTA performance in adversarial robustness, the findings contribute methodological insights and emphasize the importance of adaptive attacks in evaluating model robustness.

**Audience:**

Yes

**Broader Impact Concerns:**

None.

**Claims And Evidence:**

Yes

**Requested Changes:**

As mentioned in the weakness.
1. Please discuss or compare some relevant randomized defense.
2. Please discuss and explore the reasonable evaluation of randomized ensemble.
3. Please provide an adversarial transferability study.

**Strengths And Weaknesses:**

Strengths:
1. The proposed framework is simple and easy to follow. It includes a front-end enhanced neural model that combines a frozen backbone classifier with a convolutional model, which enhances adversarial robustness against gradient attacks.
2. The experiments are extensive. The evaluation includes multiple datasets, such as CIFAR-10/100 and ImageNet. Compared with baselines, the proposed framework achieves superior performance.
3. The discussion on the limitations of existing evaluation approaches, particularly regarding the reliability of the AutoAttack package could benefit the community.

Weaknesses:
1. There exist many randomized defense techniques, such as [a,b,c,d], which should be discussed or compared.
2. Although the discussion on the limitations of existing evaluation approaches can be valuable, more discussion of a potential reasonable evaluation of randomized ensemble could be more valuable.
3. The design of the proposed randomized ensemble is highly correlated to the adversarial transferability of selected models. The adversarial transferability study could be included to demonstrate its influence.

[a]. Defending against adversarial attacks by randomized diversification. CVPR 2019.

[b]. Certified adversarial robustness via randomized smoothing. ICML 2019.

[c]. Double-win quant: Aggressively winning robustness of quantized deep neural networks via random precision training and inference. ICML 2021.

[d]. Random normalization aggregation for adversarial defense. NeurIPS 2022.

---

> ### Author Response · Authors · 2024-04-30
> **we can expand the discussion, but some of the proposed changes, we believe, are beyond the scope of the paper**
>
> Thank you for the thoughtful comments,
>
> We will discuss additional randomization techniques, however, the study focuses on the "strength" of the AutoAttack, plus, we would like to document a case of unusually strong gradient masking, which is completely unexpected in our opinion given that the front end is differentiable and has a skip connection.
>
> > Please provide an adversarial transferability study.
>
> Please, note that we use a variant of a transfer attack to break the "defense" to demonstrate that front-end enhanced models are not fundamentally more robust compared to the backbone classifier model.
>
> > Please discuss and explore the reasonable evaluation of randomized ensemble.
> We could "strengthen" the discussion about the difficulty of such a problem, but we believe a definitive solution is not something that can be invented easily. In fact, in the section "Is AutoAttack in principle effective against randomized ensembles?" we show that without gradient masking randomized ensembles are easily "breakable" by strong gradient attacks. However, they fail when individual models mask gradients. In summary, we think this is a complicated problem that is well beyond the scope of the paper.

---

> ### Author Response · Authors · 2024-06-19
> **transfer attacks**
>
> PS: The discussion period is over, but I still would like to note that (on a second thought) I would add experiments with transfer attacks. In fact, I did some experimentation in this regard and the results were rather interesting. Transfer attacks effectiveness is limited by the transferability among models, but in some cases this transferability is quite good, so this finding / experiment can strengthen the paper.

---

### Review · Reviewer_5CEV · 2024-04-26

**Summary Of Contributions:**

This work conducted experiments on front-end enhanced neural models where a frozen backbone classifier was prepended by a differentiable and fully convolutional model with a skip connection. It was observed that the resulting models, after training, were unusually resistant to gradient attacks. The authors suspected that the reason for the observed resilience is the gradient masking phenomenon.

**Audience:**

Yes

**Broader Impact Concerns:**

No concerns on the ethical implications of this work

**Claims And Evidence:**

No

**Requested Changes:**

Besides the "Weaknesses" listed above, I have the following detailed comments:

1. Some minor consistency issues: e.g., "Croce et al. (2021)" vs. "(Croce & Hein, 2020a)".
2. The abstract is relatively longer than average, and several sentences in the abstract merely repeat content from the introduction.
3. The explanation of how prior attacks are processed is placed in the Methods section, which may be confusing to first-time readers, as it is unclear whether it is part of the preliminaries or the proposed method. Moving this introduction to prior works to the "Preliminaries" section might be a better option.

**Strengths And Weaknesses:**

# Strengths
1. Targets an important question: The robustness of deep learning models is a significant problem.
2. Provides interesting discussions based on the experimental results.
3. Includes both CNN and ViT in the experiments.

# Weakness
1. Paper organization: The organization of the paper could be improved. Specifically, the authors mix their proposed techniques with prior works in one section, making the core contribution unclear. More detailed comments are in the "Requested Changes" section.
2. Unclear contribution: It seems that the "front-end enhanced neural models, where a frozen backbone classifier is prepended by a differentiable and fully convolutional model with a skip connection," were not proposed by the authors, as the model structures follow those in DnCNN (Zhang et al., 2017) and Norouzzadeh et al. (2021). Also, the attack methods are not new, as detailed in Sec. 4.2. Based on my understanding, the authors merely modified the settings of prior attack methods to challenge the "front-end enhanced neural models." Consequently, the claim of "methodological contributions" may not be valid. Moreover, the statement "further supports the thesis..." does not sufficiently contribute because the authors have not clearly delineated what is new compared to prior works.
3. Validity of experimental settings is not clarified: The authors claim that "we use randomly selected images: 500 for CIFAR10/CIFAR100 and 200 for ImageNet", however, no prior works are cited for this setting, and no experimental results are provided to prove its validity.

---

> ### Author Response · Authors · 2024-04-30
> **we have a novel finding despite using existing "machinery"**
>
> Thank you for the thoughtful comments.
>
> It is going to be rather straightforward to address editorial suggestions.
>
> We would like to respond to the main criticism here:
>
> Although we use similar methods with only modest modifications, we achieve completely different objectives and results compared to prior work. Our contribution is an (accidental) discovery of a surprisingly strong gradient masking phenomenon, which we believe is unexpected in this setting. Given that the frontend is fully differentiable and even has a **skip-connection** we do not have a good explanation why this happens.
>
> Our contribution is **methodological** in the sense that further questions reliability of automatic evaluation approaches (i.e., contributes **to the methodology of robustness evaluation**). It is not methodologically novel from the perspective of proposing novel methods or defenses.
>
> > Validity of experimental settings is not clarified: The authors claim that "we use randomly selected images: 500 for CIFAR10/CIFAR100 and 200 for ImageNet", however, no prior works are cited for this setting, and no experimental results are provided to prove its validity.
>
> We "validate" results by computing 99% confidence intervals. For a revision we would make it more clear.

---

### Review · Reviewer_wEQx · 2024-04-28

**Summary Of Contributions:**

This submission explores the effect of using a differentiable, fully convolutional front-end layer with skip connections before a frozen backbone on gradient-based adversarial attacks. Furthermore, this submission questions the reliability of the increasingly popular AutoAttack package, which includes multiple gradient-based attacks. The authors suggest AutoAttack may overestimate the robustness of some models, specifically those that use randomized ensembles of models including the differentiable front end.

The authors found that the differentiable, fully convolutional model can seemingly preserve the classifier’s ability to make accurate predictions on clean data, while increasing robustness to gradient-based attacks. For non-gradient based attacks, a single front-end enhanced model does perform significantly worse, with almost 0% accuracy in some cases. However, when using randomized ensembles of the same front-end enhanced models, the accuracy for non-gradient-based attacks rivals that of the current state-of-the-art models. This discrepancy suggests that AutoAttack may not be as reliable as previously claimed.

The authors suggest that this may be due to an unexpected phenomenon seen in these front-end layers known as gradient masking, typically seen in “gradient shattering components” such as JPEG compression. The authors investigate and provide reasons as to why they believe gradient masking is responsible for this behavior.

**Audience:**

Yes

**Claims And Evidence:**

Yes

**Requested Changes:**

Proposed Adjustments to strengthen your work:

1. In the first sentence in the abstract, "a frozen backbone classifier was prepended by a differentiable and fully convolutional model with a skip connection" sounds as if the front-end layer comes before the classifier, not before the backbone. This is clarified later on, but I'd suggest rewriting to avoid a potential misinterpretation from the reader.

2. In the first paragraph of the introduction, standardize the references to AutoAttack to improve clarity. Currently "The AutoAttack", "AutoAttack", and "The AutoAttack package" are all used which could confuse the reader.

3. In the first sentence of the second paragraph in the Related Work section, the connection between the linear and non-linear classifiers is not well established. The sentence moves quickly from discussing linear classifiers and their susceptibility to attacks to previous beliefs that complex classifiers were more difficult to attack. Consider being more explicit in the wording of this sentence.

4. For the caption of Figure 1, I suggest changing "shortcut connection" to "skip connection" as it term used most in the paper and is used in the title.

5. For the first set of bullets in section 4.1, I would suggest being more explicit in the description of tested models. Specifically, for the second bullet, it sounds as if standard and robust are being used synonymously when the first bullet uses standard and non-robust synonymously.

6. For the final set of bullets in section 4.1, explain why the model is trained for at most one epoch.

7. In the first sentence of the paragraph discussing Zero-order PGD in section 4.2, if going to make a comparison between zero-order PGD and smooth PGD, smooth PGD should be briefly introduced before this section. I would suggest removing this line or adding a short paragraph discussing smooth PGD to establish necessary context.

8. In the third sentence of paragraph 5 in section 5.2, explain why the accuracy of the Transfer attack is used as the "true robust accuracy of this model".

9. In the "Is it gradient masking" section in section 5.2, provide more information for each of the three supporting reasons as to why the exaggerated performance is due to gradient masking. A sentence or two highlighting why these reasons could be an indicator would help solidify your rationale for a more general audience.

10. Sentence 3 of the Conclusion is copied and pasted four times throughout the paper. While this is a good way to describe the gradient masking phenomenon for differentiable models, refrain from using this exact sentence multiple times throughout the writing. You could either remove some of the instances of this sentence of rephrase them to cover the same information with different wording.

To secure my recommendation, go through the submission and correct spelling and grammar issues to the best of your ability. Proposed adjustments:

1. In the second sentence of the third paragraph in the related works section, the unnecessary use of colons negatively affects the readability of this section. Consider separating the sentences using commas or new sentences.

2. For the first bullet in the first set of bullets in section 4.1, there are too many commas being used, and the use of "i.e." seems forced. Simplify this bullet to improve readability

3. In the first sentence of the third block of text in section 4.1, correct "randomly a uniformly".

4. In the last sentence of the block of text in the AutoAttack section of section 4.2, correct "AuttoAttack"

**Strengths And Weaknesses:**

This submission thoroughly investigates its claims by evaluating the front-end enhanced models on different datasets and architectures. However, the clarity of the writing could be improved to allow the reader to follow the reading more easily. Specific examples are mentioned in the "Requested Changes" section.

---

> ### Author Response · Authors · 2024-04-30
> **clarity improvements ack**
>
> Thank you very much for a thoughtful review and suggestions!
>
> We agree that the clarity can be improved and the proposed suggestions all seem to be quite helpful. We will implement them for the final version of the paper.

---

### Decision · Action_Editor_WUiy · 2024-07-20

**Recommendation:** Accept with minor revision

**Comment:**

This paper proposed adding a denoising front end (DnCNN) architecture from Zhang et al. (2017) with a skip connection to pre-trained image classification backbones. The empirical results on multiple datasets and model architectures show that training this layer along can have the gradient masking effect that obfuscates the tested white-box attacks (in a non-adaptive setting), while it is not effective against tested non-gradient based and black-box attacks. All reviewers appreciate the extensive empirical analysis, but share similar concerns as listed below.

Based on review feedback, author response and my own reading, I suggest a mandatory revision and will accept the revised version only if all the requirements are met.

- Presentation and organization: please follow the reviewers' comments to improve the presentation and organization.
- Novelty to DnCNN: please elaborate on the difference to the original DnCNN paper (both in terms of architecture changes and the scope of the research). Also, do the claimed results hold only for DnCNN but not for other types of similar front-end architectures? How critical is the skip connection? Adding some ablation study on the importance of the proposed architecture for the claims will add more clarity.
- Experiments on transfer attacks: please add the transfer attack results, as discussed with the reviewers.

**Audience:**

Of broad interest to adversarial machine learning and security community

**Claims And Evidence:**

This paper provides empirical evidence on several models and datasets to show that adding a DnCNN layer in front of a frozen backbone can be effective in causing gradient masking. Combined with other defenses, it can be used to improve the robust accuracy against adversairl examples.

**Resubmission Of Major Revision:**

The authors may consider submitting a major revision at a later time.

---

> ### Author Response · Authors · 2024-08-08
> **question about a major revision**
>
> Thank you very much!
>
> >The authors may consider submitting a major revision at a later time.
>
> We lean towards adding these experiments. Does "at a later time" mean that we have more time to do a revision?

---

> > ### Comment · Action_Editor_WUiy · 2024-08-08
> > **Yes**
> >
> > Dear Authors,
> >
> > I don't think TMLR has a specific deadline for finishing the revision. So please take your time and ensure the necessary changes will be made.